# *XAANTAL1* Reveals an Additional Level of Flowering Regulation in the Shoot Apical Meristem in Response to Light and Increased Temperature in Arabidopsis

**DOI:** 10.3390/ijms241612773

**Published:** 2023-08-14

**Authors:** Mónica Rodríguez-Bolaños, Tania Martínez, Saray Juárez, Stella Quiroz, Andrea Domínguez, Adriana Garay-Arroyo, María de la Paz Sanchez, Elena R. Álvarez-Buylla, Berenice García-Ponce

**Affiliations:** 1Instituto de Ecologίa, Departamento de Ecologίa Funcional, Universidad Nacional Autónoma de México, Circuito ext. s/no. Ciudad Universitaria, Coyoacán 04510, CDMX, Mexico; 2Laboratory of Pathogens and Host Immunity, University of Montpellier, 34 090 Montpellier, France

**Keywords:** flowering, XAANTAL1, FD, long-day photoperiod, PHYB, high permissible temperature

## Abstract

Light and photoperiod are environmental signals that regulate flowering transition. In plants like *Arabidopsis thaliana*, this regulation relies on CONSTANS, a transcription factor that is negatively posttranslational regulated by phytochrome B during the morning, while it is stabilized by PHYA and cryptochromes 1/2 at the end of daylight hours. CO induces the expression of *FT*, whose protein travels from the leaves to the apical meristem, where it binds to FD to regulate some flowering genes. Although PHYB delays flowering, we show that light and PHYB positively regulate *XAANTAL1* and other flowering genes in the shoot apices. Also, the genetic data indicate that XAL1 and FD participate in the same signaling pathway in flowering promotion when plants are grown under a long-day photoperiod at 22 °C. By contrast, XAL1 functions independently of FD or PIF4 to induce flowering at higher temperatures (27 °C), even under long days. Furthermore, XAL1 directly binds to *FD*, *SOC1*, *LFY*, and *AP1* promoters. Our findings lead us to propose that light and temperature influence the floral network at the meristem level in a partially independent way of the signaling generated from the leaves.

## 1. Introduction

Environmental and plant endogenous signals regulate the flowering transition, a developmental process essential for angiosperm reproduction and seed viability [1]. For plants such as *Arabidopsis thaliana* that inhabit latitudes far from the equator, light sensing, day length (or photoperiod), and temperature are relevant signals that allow them to bloom in spring–summer seasons when there are long days (LD) and temperature increases.

To match the flowering time with optimal environmental conditions, LD and the rise in temperature signals converged into a complex multilevel regulatory network [2]. Perception of the LD photoperiod relies mainly on the accumulation of CONSTANS (CO), a zinc-fingered B-box transcriptional factor [3] regulated at the transcriptional and posttranslational levels. On the one hand, *CO* transcript levels cycle in a circadian manner, with maximum accumulation at 16 h after light perception that is maintained during the night and decreases during the day [4]. Several proteins, including the CYCLING DOF FACTORs (CDFs) repressors, and the F-box E3-ubiquitin ligases FLAVIN BINDING, KELCH REPEAT, F-BOX1 (FKF1), and ZEITLUPE (ZTL) photoreceptors are responsible for *CO* mRNA cycling. FKF1, in association with GIGANTEA (GI), targets CDFs for their degradation, while GI can induce *CO* and *FLOWERING LOCUS T* (*FT*) directly [5,6,7,8]. On the other hand, CO protein accumulates at dusk and is degraded at night [9]. The E3-ligase complex formed by CONSTITUTIVE PHOTOMORPHOGENIC 1 (COP1) and SUPPRESSOR OF PHYTOCHROME A-105 (SPA1) ubiquitinate CO for degradation by the proteasome [10,11]. However, the photoreceptors PHYTOCHROME A (PHYA) and CRYPTOCHROME 2 (CRY2) inhibit the action of COP1-SPA1 at the end of daylight [12,13]. This allows CO to activate *FT* in the phloem companion cells under LD but not on short days (SD) [14]. Although *CO* is widely expressed, it was demonstrated that *CO* expression by phloem-specific promoters is sufficient to complement the *co* mutant and to induce early flowering but not when expressed by meristem-specific promoters [15].

Photoreceptors are activated by different light spectrum wavelengths and are involved in many plants’ physiological and developmental processes [16]. Phytochrome B (PHYB) is activated by red light, while PHYA and CRY1/2 are activated by far-red and blue light, respectively, and all of them can accumulate in the nucleus [17]. PHYB, unlike the other two photoreceptors, promotes CO degradation in the morning, probably mediated by HIGH EXPRESSION OF OSMOTICALLY RESPONSIVE GENE1 (HOS1) [9,18,19]. In congruence with the regulation of these photoreceptors on CO abundance and, thus, *FT* expression, the *phyB* mutant is early flowering compared to wild-type plants in LD, and the *phyA* and *cry1 cry2* mutants delay flowering of *CO* overexpression [9].

PHYB also acts as a thermosensor [20]. At higher temperatures, PHYB undergoes a phenomenon called thermal reversion in which its active form, Pfr, changes to its inactive isoform, Pr, in a light-independent reaction [2,21]. Under these circumstances, there is less Pfr-mediated degradation of CO and PHYTOCHROME INTERACTING FACTOR 4 (PIF4) [22,23]. PIF4 is a positive regulator of *FT* in response to permissible elevated temperatures, independently of the LD photoperiod. However, *FT*’s induction increases when CO interacts with PIF4 [24]. Accordingly, *pif4* is a late flowering mutant when growing under SD at 27 °C [25], while the expression of *FT* increases in leaves of the *phyB* mutant or when PHYB Pfr is inactivated by thermal reversion [22,25].

*FT* is translated in the leaves from which it travels systemically in association with FT-INTERACTING PROTEIN1 (FTIP1) and SODIUM POTASSIUM ROOT DEFECTIVE 1 (NaKR1) to the apical meristem [1,26,27,28,29]. FT is a small 23 kDa protein that must associate with FD, a bZIP-like transcription factor expressed in the apical meristem, to carry out its activity as a florigen [30]. Thus, FT functions as a transcriptional cofactor of FD to induce the expression of the MADS-box factors *SUPPRESSOR OF OVEREXPRESSION OF CO* (*SOC1*), *FRUITFULL* (*FUL*), and *APETALA1* (*AP1*), and members of the *SQUAMOSA BINDING PROTEIN LIKE* (*SPL*) family, *SPL3/4/5* genes [14,30,31,32,33,34]. In turn, SPLs facilitate FT/FD binding to induce *LEAFY* (*LFY*) and *AP1* [33], which give identity to the flower meristem [35]. LFY directly upregulates *FD* expression [36]. However, it has been shown recently that FT-FD association must be transient as if it persists, it is detrimental to flower formation. One way to prevent FT-FD formation is to inhibit *FD* expression in the flower meristem by AP1 [37,38].

Even though temperature increases as day length during the spring-summer seasons, there is evidence that photoperiod and temperature may induce flowering via independent pathways. For example, the flowering time of the *gi* and *co* mutants is accelerated by elevated temperatures [39]. Although, the *ft-10* mutant was found unable to flower in response to increased temperature, indicating that FT is required to induce flowering in response to both the LD photoperiod and high temperatures [39]. However, under the latter condition, PIF4 and PIF5 directly induce *FT* expression independently of CO [25]. Interestingly, the binding of PIF4 to the *FT* promoter is favored at 27 °C, possibly because of nucleosome-H2A.Z turnover that allows chromatin relaxation at this temperature [25]. Nevertheless, the relationship between these two signals seems to be more complex than parallel pathways, as the gain of function of CRY2 in the Cabo Verde Islands accession (Cvi-0) causes these plants to flower early, although they are less sensitive to elevated temperatures [40]. Also, PHYB, COP1, and PIFs are common factors that regulate flowering and plants’ growth in response to both signals [16], suggesting that they are part of a complex network.

Although the flowering induction pathway in response to the LD photoperiod has been extensively studied, other factors must be implicated in flowering transition in response to the LD photoperiod since the triple mutant *ft soc1 lfy* can still flower [41](). Noteworthily, *FD* accumulation during flowering at the apical meristem is independent of FT activity [42], but finding positive regulators of *FD* other than LFY has remained elusive. Also, there is little knowledge of the components of the flowering network in response to high temperatures besides *FT* induction by this condition.

*XAANTAL1* (*XAL1/AGL12*) is a MADS-box gene implicated in flowering transition and root development. Mutant plants in this gene are late flowering compared to WT when grown under the LD photoperiod but do not show any phenotype when cultivated under SD or after vernalization treatment [43]. However, genetic relationships with other members of the flowering network were not yet known.

In this work, we performed a *XAL1* expression analysis and established the genetic interactions with other members of the flowering network in response to the long-day photoperiod and increased temperature. Our data indicate that *XAL1* is upregulated by light and, together with other members of the flowering network expressed in the apical meristem, they are positively regulated by PHYB, despite its negative effect on CO and PIF4. We also found that *XAL1* and *FD* may have epistatic or independent relationships depending on the photoperiod and temperature used. Moreover, XAL1 directly binds to the *FD* promoter and other key flowering genes, such as *SOC1*, *LFY* and *AP1*. Overall, our results allow us to propose that the regulation of flowering, in response to LD photoperiod and temperature at the apical meristem, is partially independent of the regulation given by CO and PIF4, adding another level of control to fine-tune FT-derived signaling.

## 2. Results

### 2.1. Genetic Relationships of XAL1 with Other Members of the Photoperiod Pathway

To find out possible genetic relationships between *XAL1* and other members of the photoperiod pathway, we selected three different double mutants, *co-1 xal1-2*, *xal1-2 fd-3*, and *xal1-2 soc1-6*, and we analyzed their flowering time under a LD photoperiod (Figure 1A–C). The double mutants *co-1 xal1-2* and *xal1-2 soc1-6* have an additive late-flowering phenotype compared to the single mutants, suggesting that CO and SOC1 function independently of XAL1. We could not obtain the *xal1 ft-10* double mutant after analyzing 36 plants in the F2 and 50 plants F3 descendants from a double heterozygous plant, suggesting the double homozygous line is lethal. Since FT partners with FD at the inflorescence meristem [30,31], we generated the *xal1-2 fd-3* double mutant. In this case, *xal1-2 fd-3* showed a similar flowering time to *fd-3*; therefore, *FD* is epistatic over *XAL1* (Figure 1B). In congruence with these results, we found that *XAL1* accumulates mainly in the shoot apex compared to the leaves of 14-day-old wild-type (WT) plants (Figure 1D). As a negative control for possible contamination of the apex samples with vegetative tissue, we used the *FT* expression, which accumulates only in the leaves of these samples (Appendix A). Moreover, the expression analysis of *XAL1* and *FD* in the mutant apices of the other gene showed that XAL1 is a positive regulator of *FD* and not vice versa (Figure 1E,F). This is relevant since there is little information regarding positive regulators of *FD*.

When we analyzed the expression of different flowering genes in apices of *xal1-2* compared to WT plants, we found that XAL1 is a positive regulator of the MADS-box *SOC1*, *AGL24*, *FUL*, and *AP1*, some members of the *SPL* family *SPL3/4/5* and *SPL9/15*, and *LFY* (Figure 2 and Appendix A). To confirm that XAL1 and FD function in the same regulatory pathway, we chose some representative flowering genes and analyzed their expression in the *xal1-2 fd-3* double mutant compared to the single mutants and the WT. We observed that all genes have similar lower accumulation levels in apices of single mutants compared to the WT (Figure 2). Moreover, *SOC1*, *FUL*, *AP1*, and *SPL9* have similar accumulation in the *xal1-2 fd-3* double mutant, compared to either one or both single mutants, suggesting that XAL1 and FD act in the same pathway. *SPL3* was the only gene analyzed that showed a significant decrease in the double mutant compared to the single parentals, suggesting that XAL1 and FD may regulate some genes independently. Intriguingly, we found that *LFY* and *SPL4* significantly increased in the double mutant relative to the single mutants, suggesting that in the absence of XAL1 and FD other mechanisms induce those genes probably because of the absence of negative feedback mechanisms. The results show that XAL1 and FD may function in the same or converging pathways to regulate the flowering genes at the apical meristems.

### 2.2. XAL1 Binds to the Promoters Regions of FD, SOC1, LFY, and AP1

To establish whether XAL1 directly or indirectly regulates *FD*, *SOC1*, *LFY*, and *AP1*, we first generate constitutive overexpression lines (OE) of XAL1 carrying a GFP tag (*35S::XAL1_cDNA_::GFP*) or without GFP (*35S::XAL1_cDNA_*) transformed into the *xal1-2* background. Both lines were selected for their ability to complement the short-root phenotype of *xal1-2* (Appendix A; ref. [43]), indicating that XAL1 is functional. We also confirmed the presence of GFP in the labelled OE line (Appendix A). However, when we analyzed the flowering time of both lines, we found that only the complementation plants carrying the GFP were slightly early flowering compared to the WT (Appendix A). Since the *xal1-2* mutant has a subtle but reproducible late flowering phenotype [43], and its function was confirmed by the increased delay in the double mutants observed in Figure 1, the results suggest that XAL1 is necessary but not sufficient for a strong early flowering induction. Consistent with these results, we found that only two of the analyzed genes changed their expression pattern in the *35S::XAL1::GFP/xal1-2* compared to the WT in a three-time kinetics: *FD* has higher accumulation levels than the WT in 14-day-old plants, and *AP1* is prematurely induced compared to WT plants at 10 days (Appendix A). As expected, *XAL1* was highly overexpressed during the kinetics compared to the WT, and we also confirmed the semiquantitative RT-PCR result using RT-qPCR in 14-day-old plants (Appendix A).

Chromatin immunoprecipitation (ChIP) assays were performed using an anti-GFP antibody with DNA extracted from both complementation lines, and the WT was included as an additional negative control (Figure 3). We chose four key floral genes, *FD*, *SOC1*, *LFY*, and *AP1*, positively regulated by XAL1 under LD conditions. Three to four DNA fragments were amplified after ChIP experiments with primers flaking canonical and noncanonical CArG boxes, mainly in the intergenic region upstream of the ATG codon. Some of the CArG motifs have been previously reported [42,44,45,46,47], and some others were found using different motif search databases such as PLACE, PlantPAN, Athamap, and Meme Suite 5.5.3 [48,49,50,51,52,53].

In the case of *FD*, three fragments were enriched from the immunoprecipitated DNA (Figure 3). The motif CCAAATGTGG (CC[W]_4_NNGG) in fragment I is identical to the CArG-box sequence reported as an AGL15 binding site to its own promoter [46]; the CTAATTTTAG and CATATAATAG (C[W]_8_G) sequences in fragments II and IV, respectively, were reported as FLOWERING LOCUS C (FLC) binding sites to *FD* [42]. For the *SOC1* promoter, the most substantial enrichment was found on fragment III, which includes two C[W]_7_G boxes (CTATTTTTG and CAAAATAAG). Interestingly, the second one is particularly relevant for *SOC1* repression by AP1-SOC1, SEPALLATA 3 (SEP3), and FLC [42,47]. Also, the CArG-box (CCAATATAG) in fragment II is important for *SOC1* induction by AGL24 [54]. We also recovered three enriched fragments from the 5′ upstream region of the *LFY* gene. Fragment I is like the one amplified by Liu and collaborators (2008) [54], in which they observed the binding of SOC1 to *LFY*. Fragment II includes the CArG-box (CATATATGG), and fragment III contains at least two CArG-boxes (CAAAAATAGC and CAATAAAG). Finally, we also found two enriched fragments for *AP1*. Fragment I (CAATATATG), which is farther from the ATG, showed stronger accumulation than fragment IV (CCATTTTTGG), which includes one of the CArG-boxes found in the first intron.

Our results suggest that *FD*, *SOC1*, *LFY*, and *AP1* genes are direct targets of XAL1. However, XAL1 may require associating with other factors to activate their expression since they were not highly increased in the *XAL1* OE-lines. Further studies are needed to establish the relevance of each CArG-box in the regulation of those genes by XAL1.

### 2.3. XAL1 Is Induced by Light Probably via PHYB at the Shoot Apex

Considering that XAL1 participates in the photoperiod pathway independently of CO and positively regulates *FD*, we wondered whether *XAL1* could be regulated by light or the circadian clock at the shoot apex. Notably, 24-h kinetics showed an increase in *XAL1* expression after two hours of light exposure compared to time zero (15 minutes before the lamps turned on), and slowly decreased to basal accumulation levels after 16–24 h (Figure 4A). This expression pattern is opposite to the one observed for genes such as *GI*, *CO*, and *FT* regulated by the circadian clock in the leaves [4,55]. Since *XAL1* mRNA levels peak shortly after the lamps are turned on, this led us to investigate if *XAL1* expression is regulated by light at the apical meristem. The plants were grown under a long-day photoperiod for thirteen days; they were kept in complete darkness for two days and then exposed to light again. Apices samples were collected every two hours after this procedure. *XAL*1 accumulation levels peak between 2 and 4 h relative to plants at time zero (Figure 4B), and the basal expression level did not change when plants were kept in the dark for ten hours. Moreover, *XAL1* accumulation quickly decreased when plants were grown for eight hours in the light and then transferred to the dark for two hours. These results strongly suggest that *XAL1* is regulated by light.

Phytochromes and cryptochromes are photoreceptors relevant for all photomorphogenic processes. Among them, it is known that PHYB, PHYA, and CRY1/2 have essential roles in regulating CO protein stability during the flowering transition [9,13]. Interestingly, we found that *XAL1* expression decreased in *phyB* apices at 14 and 18 days after sowing (DAS) (Figure 4C), while it did not change in the mutant leaves compared to wild-type plants (Figure 4D). We did not find any difference in *PHYB* expression between leaves and apices of 14-day-old plants (Figure 4E), suggesting that the *XAL1* expression difference observed in the shoot apex is not due to reducing PHYB levels at the meristem compared to leaves. Additionally, no significant change in *XAL1* expression was observed in *phyA*, *cry1* and *cry2-1* mutant apices compared to WT (Appendix A). Thus, *XAL1* is specifically and positively regulated by PHYB at the apex.

PHYB negatively regulates CO in the leaves. Therefore, in the absence of this photoreceptor, CO and FT accumulate and, consequently, *phyB* plants are early flowering [22]. Contrary to this notion, *XAL1* is reduced in *phyB* plants. Hence, we wondered if this effect was extended to other flowering genes expressed at the shoot apex. We analyzed the expression of known flowering genes regulated by FD/FT signaling [33,34], including *FD* (Figure 5A–C). Unexpectedly, we observed lower *FD*, *SOC1*, *FUL*, and *LFY* transcript levels in *phyB* compared to WT apices (Figure 5A). *SPL3* and *SPL15* did not change between the *phyB* plants and the WT (Figure 5B), and only *SPL4*, *SPL9*, and *AP1* showed an increased accumulation in the mutant compared to the WT (Figure 5C). We confirmed that *FT* and the *TWEEN SISTER OF FT* (*TSF*) were induced in the leaves of the same *phyB* plants used for apices samples (Figure 5D). Therefore, the results suggest that the early flowering phenotype of *phyB* mutant relies on some, but not all, of the flowering genes known to be regulated by FD/FT.

In agreement with those expression results, when we quantified the flowering phenotype of the double mutant *xal1-2 phyB* compared to the single mutants and the WT, we found that it behaves as a late flowering mutant like *xal1-2*, even though *phyB* is early flowering (Figure 5E). The results imply that PHYB functions in the leaves and the apical meristems are partially independent, and the flowering gene regulatory network at the apical meristem can counteract FT induction in the *phyB* mutant.

### 2.4. XAL1 and FD Independently Regulate Flowering at High Temperatures

High permissible temperatures induce flowering under short-day conditions indicating that temperature and photoperiod signals may act independently. Nevertheless, FT, an indispensable signal in response to the LD photoperiod, also plays an important role as a flowering signal when flowering is induced under SD at 27 °C [25,39]. However, there is no information regarding the effect of this environmental condition on *FD* or other genes in the apical meristem.

Considering that XAL1 functions in the same flowering pathway as FD under LD at 22 °C, we decided to analyze the impact of warm temperatures on the flowering phenotypes of *xal1-2* and *fd-3* mutants when they were grown under SD and LD photoperiods. Under SD at 27 °C, *xal1-2* shows a strong late flowering phenotype compared to the WT and similar to *fd-3* (Figure 6A,B). Notably, the double *xal1-2 fd-3* mutant had an additive phenotype compared to the single mutants. Hence, contrary to the epistatic relationship observed when plants were grown under LD at 22 °C, XAL1 and FD participate independently in the induction of flowering in response to higher temperatures.

Since in the summer season the LD photoperiod and high temperatures coincide, we wondered if any of those signals has a prevalent effect when plants are grown under LD at 27 °C. We found that photoperiod and high temperature are additive, accelerating flowering time even more than when plants are grown under SD at 27 °C (Figure 6B,C) or LD at 22 °C (Figure 1B). However, under the LD photoperiod at 27 °C, *fd-3* flowered later than *xal1*, which behaves almost like the WT. Nevertheless, the double mutant maintains its additive phenotype compared to the parentals (Figure 6C). These results indicate that XAL1 and FD function independently under high temperatures to induce flowering regardless of the photoperiod. Also, the data suggest that elevated temperatures may modify the genetic interaction between *FD* and *XAL1*, observed under LD at 22 °C. Control plants grown under SD at 22 °C showed no significant difference in their flowering time compared to the WT, as expected (Appendix A).

It has been established that PIF4 induces *FT* in the leaves in response to 27 °C under SD, which triggers the flowering transition at the inflorescence meristem [25]. Our results showed that XAL1 and FD also induce flowering under this growth condition. To know the relation between *XAL1* and *PIF4*, we generate the double mutant *xal1-2 pif4-2* and analyze its flowering phenotype compared to single mutants and the WT grown under SD at 27 °C (Figure 6D). The *xal1-2* mutant shows a later flowering phenotype than *pif4-2* compared to the WT, and no significant difference among genotypes was detected under SD at 22 °C (Appendix A). Furthermore, the double mutant has a strong additive phenotype compared to the single mutants and the WT plants, indicating that XAL1 and PIF4 induce flowering in response to warm temperatures independently.

We corroborated that high-temperature treatment induces relevant flowering genes, like *XAL1*, *FD*, *PIF4*, *LFY*, and *AP1*, in WT plants grown under SD at 27 °C against 22 °C (Appendix A). Then, we analyzed the expression of several genes in apices of WT and *xal1-2* plants subjected to 27 °C (Figure 6E,F). Interestingly, *FD*, *SOC1*, *FUL*, *AGL24*, *SPL3*, and *LFY* decreased in the mutant compared to the WT, suggesting that XAL1 positively regulates them under this environmental condition. *SPL9* and *SPL15* expression did not statistically change between genotypes. However, *SPL4* and particularly *AP1* mRNA levels increased in the *xal1-2* mutant. Notably, the latter genes also have higher accumulation levels in the *phyB* mutant when it was grown in LD at 22 °C (Figure 5C), supporting the idea that PHYB Pfr inactivation by thermal reversion leads to the induction of *FT* and consequently *SPL4* and *AP1* [39]. Also, our results indicate that XAL1 favors the repression of these two genes at warm temperatures, contrary to what we found in the LD photoperiod (Figure 2).

## 3. Discussion

Light quality, photoperiod, and temperature are environmental signals that impact plant development processes, including flowering transition [16,56]. In this work, we analyzed the involvement of XAL1 in the regulation of flowering in response to the LD photoperiod, as well as the effect of increasing the temperature to 27 °C. Notably, XAL1 regulates *FD* and other flowering genes at the apical meristem under LD. Also, we found that the genetic relationship between *XAL1* and *FD* changes depending on the environmental condition; *FD* is epistatic over *XAL1* under LD, but they function independently to induce flowering in response to elevated temperatures. This is noteworthy because FD had not been implicated in the latter condition. Finally, we showed that *XAL1* is positively regulated by light, probably via PHYB, as well as other flowering genes. Therefore, the analysis presented here on XAL1 led us to uncover new functions of PHYB and FD during the flowering transition at the apical meristem.

### 3.1. XAL1 Involvement in the Flowering Network Control in Response to LD Photoperiod and Increased Temperature

The *xal1* mutant was characterized as a late flowering mutant under the long-day photoperiod, and it was shown that on short days or after vernalization treatment the mutant was not different from the WT, indicating that XAL1 participates in the induction of flowering under LD [43].

The genetic analysis of *XAL1* with different members of the LD photoperiod pathway showed that it participates in the same pathway as *FD* and independently of *CO*. These results, together with the expression analysis showing that *XAL1* transcript accumulates more in apices of soil growing plants compared to its levels observed in leaves, support the idea that the role of XAL1 in flowering is centered at the apical meristem. Accordingly, previous in situ hybridizations showed that *XAL1* is expressed in the inflorescence and the flower meristems [43].

Interestingly, *XAL1* levels increased after short periods of light induction and decayed slowly throughout the day. This expression pattern is opposite to that observed for those genes involved in the long-day photoperiod pathway regulated by the circadian cycle, such as *GI*, *CO*, and *FT* [4,7,55]. We also showed that PHYB positively regulates *XAL1* accumulation while PHYA and CRY1/2 do not have any impact when plants are grown under white light. Therefore, it seems that *XAL1* and probably other genes from the network respond to light signaling at the apical meristem.

Downstream of XAL1, it positively regulates all genes analyzed from the MADS-box and SPL families and *LFY*. Moreover, except for *SPL3*, which is regulated independently by XAL1 and FD under a LD photoperiod, all other genes seem to be similarly regulated by both. These results support the genetic result that XAL1 and FD participate in the same flowering pathway in response to this environmental condition.

Until now, LFY was the only known direct inducer of *FD*; however, *LFY* is specifically expressed in the flower meristem and this regulation is transient since AP1 represses *FD* expression in this meristem [36,37,38]. Therefore, there must be other *FD* regulators in the inflorescence meristem that positively regulate it during the transition to flowering, one of them could be XAL1. Since XAL1 directly regulates *FD*, one possibility is that the regulation of the other flowering genes by XAL1 was indirect. However, we found that XAL1 can bind to the promoters of three essential genes in this process, namely *SOC1*, *LFY*, and *AP1*. We obtained several ChIP-enriched DNA fragments, including CArG motifs described as binding sites for other MADS-domain factors. However, in the case of *FD*, those motifs in the promoter have been described as FLC binding motifs to repress its expression [42]. Thus, it is possible that the occupancy of some motifs in *FD* by FLC during the vegetative state can be substituted by XAL1 and probably other transcription factors to activate it during flowering.

Despite the effect of XAL1 on the regulation of genes relevant to flowering, the *xal1-2* mutant has a slightly late-flowering phenotype in LD compared to WT, which contrasts with the strong phenotypes observed in *co-1*, *fd-3*, and *soc1-6* mutants (Figure 1). One possibility is that deregulation at the transcriptional level is not proportionally represented at the protein level. Alternatively, it is also possible that the function of XAL1 is to fine-tune the flowering network at the meristem level in response to light as an additional input signal. Further studies need to be done to clarify the XAL1 function.

Additionally, overexpression of *XAL1* is sufficient to complement the *xal1-2* root and flowering phenotypes but does not induce a very early flowering. This is probably because, besides *FD* and *AP1*, high levels of *XAL1* do not change the transcript accumulation of any of the other genes analyzed. Hence, XAL1 may require additional proteins to activate its targets. Two-hybrid analysis indicates that XAL1 may associate with AGL16, AGL21, SEP1, SOC1, and AGL74 among the MADS-domain factors [57].

We found that XAL1 and FD are involved in flowering promotion in response to high permissible temperatures. However, the genetic analysis showed that at 27 °C both genes function independently in both SD and LD photoperiods, in contrast to what was observed in LD at 22 °C. After expression analysis of several flowering genes under SD at 27 °C we found that, similarly to LD, XAL1 positively regulates *FD*, *SOC1*, *FUL*, *AGL24*, *LFY*, and *SPL3* genes; but, in contrast to the latter condition, *SPL4* and particularly *AP1* were induced at 27 °C, which can be explained by thermal reversion of PHYB Pfr (see next section). PIF4 induces *FT* in the leaves under SD at 27 °C [25]. Here, we found that XAL1 and PIF4 function independently to induce flowering, which agrees with XAL1 participation at the apical meristem. It is noteworthy that *xal1-2* showed a much stronger phenotype under SD at 27 °C than under LD at both temperatures, suggesting that the role of XAL1 at higher temperatures can be masked under LD. Therefore, our results suggest that the flowering response to light and temperature can be regulated via XAL1 at the apical meristem, although it seems it happens via independent mechanisms.

### 3.2. PHYB Regulation in Flowering Transition

PHYB is possibly involved in all photomorphogenetic and light-induced physiological processes in plants [58]. However, the knowledge about the functional role of PHYB has been centered in vegetative tissues such as leaves and hypocotyl, while its function in meristems is almost unknown.

PHYB can be in its inactive form Pr or its active form Pfr. Red light induces Pfr and far-red, or even darkness, is sufficient to revert it to its inactive Pr isoform [59]. During thermal reversion, higher temperatures accelerate the conversion of Pfr to Pr and the formation of Pfr–Pr heterodimers that are more unstable [60]. Therefore, PHYB is a photoreceptor and a thermosensor [20,21].

Pfr is preferentially found in subnuclear domains called photobodies [61,62,63], and some proteins, such as PHOTOPERIODIC CONTROL OF HYPOCOTYL1 (PCH1), promote Pfr stability even in the dark [64,65]. Moreover, Pfr binding to PIF4, leading to its degradation, is inhibited in the *pch1* mutant, which shows increased *PIF4* accumulation levels at night. Interestingly, *pch1* has an early flowering phenotype when grown in LD [23,64].

In addition to promoting protein degradation, PHYB functions at the transcriptional and posttranscriptional levels. For example, PHYB has been shown to bind to promoters of target genes and can recruit or modify the action of other transcription factors, probably affecting the regulatory dynamics of those genes. Furthermore, the association of PHYB with those regulatory regions is temperature-dependent [21]. PHYB also affects the alternative splicing of genes, including *PIFs* [66].

In flowering, PHYB is known to decrease CO and PIF4 stability, which inhibits *FT* expression [9,25,67]. On the one hand, PHYB can lead to the degradation of CO in the presence of light, possibly via HOS1 or ZTL [18,68]. On the other hand, phosphorylation of Ser-86 accelerates the inactivation of Pfr via dark or thermal reversion, which attenuates the effect of COP1 on CO, allowing its accumulation and the activation of *FT* [11,69,70]. The specific mode of action of PHYB is still unknown. However, it has been found that PHYTOCHROME-DEPENDENT LATE-FLOWERING (PHL) can counteract the inhibitory effect of PHYB in flowering transition by protecting CO from degradation [71].

In this study, we corroborate that the *phyB* mutant is indeed early flowering and that the absence of this receptor induces both *FT* and *TSF* expression in the leaves. Those proteins were expected to travel to the apical meristem and induce the flowering genes that have already been shown to be regulated by FT-FD in the meristem [30,31,33,34]. However, we unexpectedly found that at the apex many of the genes involved in flowering, including *XAL1* and *FD*, were downregulated in the *phyB* mutant, indicating that PHYB can function as a positive regulator at the apical meristem. Similarly, there was a decrease in the accumulation of these transcripts in the *xal1-2* mutant when plants were grown under SD and subjected to 27 °C.

Notably, there were two exceptions; *SPL4* and *AP1* expression levels increased in both the *xal1-2* mutant grown under SD at 27 °C and in the *phyB* mutant grown under LD compared to each WT, respectively (Figure 5 and Figure 6). Thus, it seems that upregulation of *FT* when PHYB is missing or inactivated by thermal reversion induces the expression of some flowering genes, such as *SPL4* and *AP1*, but not all. In agreement with this, the *xal1-2* mutant can counteract the early flowering phenotype of *phyB* since the double mutant *xal1-2 phyB* flowers as late as *xal1-2*. One possible explanation is that in the absence of XAL1, downregulation of *FD* affects FT-FD dimmer formation and this, combined with lower expression of other flowering genes positively regulated by FT-FD, PHYB, and XAL1 at the apical meristem, is sufficient to prevent flowering, even when *AP1* increases in the absence of PHYB. In the future, it will be relevant to study the mechanism by which PHYB positively regulates *XAL1* and the other genes.

## 4. Materials and Methods

### 4.1. Mutant Lines

*Arabidopsis thaliana* wild-type and mutant plants used in this study were in Col-0 background, except for *cry2-1*, which was in Col-4. The mutants *xal1-2* (N429367; [43]), *fd-3* (N678498), *ft-10* (N9869; [14]), *soc1-6* (SALK_138131; [72]), *phyA* (N661576), *phyB* (N675665), *pif4-2* (N879261; [73]), and *cry1* (N408687) are T-DNA insertion lines; while, *cry2-1* (N3732; [74]) and *co-1* [75] are mutants generated by fast neutrons and X-rays, respectively. Most of the seeds were provided by the Nottingham Arabidopsis Stock Centre (NASC). We obtained homozygous *co-1 xal11-2*, *xal11-2 soc1-6*, *xal11-2 fd-3*, *xal11-2 phyb*, and *xal11-2 pif4-2* double mutants selected by their sulfadiazine resistance and genotyping using oligonucleotide primers listed in Appendix A.

### 4.2. Flowering Measurements and Light Induction Experiments

For flowering analysis, plants were grown on Sunshine mix 3 (Sun Gro) soil under long-day (LD; 16 h light/8 h dark) or short-day (SD; 8 h light/16 h dark) photoperiods, with standard light intensity (120 or 240 µE m^−2^ s^−1^). We used temperatures of 22 °C or 27 °C, depending on the experiment. Flowering was assessed by measuring the bolting time in days after seed sowing (DAS) required for the stem to acquire 1 cm long, and the rosette leaf number at bolting.

For light induction experiments, plants were grown for 13 days under LD. Subsequently, they were transferred to darkness for 48 h and then returned to light exposure until the material was collected.

### 4.3. Expression Analysis

Each independent biological replicate consisting of 27–35 shoot apices or 10 leaves, respectively, was collected from 14-day-old soil-growing plants, unless indicated otherwise. RNA extraction from frozen material (150–175 mg) was carried out using Trizol reagent (Thermo Fisher, Waltham, MA, USA) followed by the Direct-zol RNA Miniprep kit (Zymo Research, Irvine, CA, USA). A total of 2 µg of RNA were reverse transcribed using the Superscript III enzyme (Invitrogen, Waltham, MA, USA; Thermo Fisher Scientific) to produce two independent cDNAs. For quantitative PCR (qPCR), three technical replicates were performed for each reaction using Maxima SYBR Green qPCR master mix 2X (Thermo Scientific) in a StepOne real-time PCR system (Applied Biosystems, Waltham, MA, USA). Data were analyzed as described in [76]. *PDF2*, *RNAH*, and *UPL7* were used as constitutive internal controls [77]. Specific semiquantitative PCR and qPCR primers are listed in Appendix A [43,54,73,75,77,78].

### 4.4. Plasmid Constructs and Transgenic Plants Selection

To generate the *35S::XAL1* (without a tag), the full-length *XAL1* cDNA was amplified using the 5′-ATTGGATCCATGGCTCGTGGAAAGATTCAGC-3′ and 5′-TTAGGATCC CTAGAACTGAAATATTTCACTTG-3′ primers that contain the BamH1 restriction sites. DNA was cloned into the p-GEM-T Easy vector (Promega, Madison, WI, USA) and, after excision, it was subcloned into the pBIN-JIT vector carrying a double 35S promoter. The correct clone orientation was confirmed by sequencing [79]. For the *35S::XAL1::GFP* construct, the *XAL1* cDNA (without the stop codon) was obtained using the *XAL1*-F 5′-ATGGCTCGTGGAAAGATTCAGC-3′ and the *XAL1*-R 5′-GAACTGAAATATTTCACTTGGCA-3′ primers. The resulting 633 bp DNA fragment was cloned into the pCR8/GW/TOPO-TA vector (Invitrogen, Thermo Fisher Scientific) and verified via sequencing. Subsequently, it was recombined into the overexpression vector pGWB5 which carries the 35S promotor and the *GFP* gene using the Gateway LR system (Invitrogen, Thermo Fisher; [80]). Both constructs were introduced into *Agrobacterium tumefaciens* (C58) and then transformed into the *xal1-2* plants using the floral-dip method [81]. Kanamycin resistant plants were selected on MS 0.2x plates (MP biomedicals, Santa Ana, CA, USA) and T2 transgenic lines were chosen based on the complementation of the *xal1-2* short root phenotype grown on sulfadiazine (20 mM) plates.

### 4.5. Chromatin Immunoprecipitation Assays

Chromatin Immunoprecipitation (ChIP) protocol was performed as described by [82]. Briefly, the wild-type, *35S::XAL1/xal1-2*, and *35S::XAL1-GFP/xal1-2* lines were grown on MS 0.2x plates for ten days and the material (1.0 g) was fixed. Chromatin was solubilized using a sonicator (Fisher Scientific; power 20%) in 20 cycles of pulses 15 s each. Immunoprecipitation was conducted overnight using an anti-GFP rabbit IgG fraction (A11122, Invitrogen; 1:5000 dilution) with A/G agarose beads (Thermo Fisher). Template ChIP DNA was amplified using primer pairs designed within or in the flanking regions of CArG boxes found primarily in the 5′ intergenic region of the *FD*, *SOC1*, *LFY*, and *AP1* genes (Appendix A). *ACTIN 2* was used as a constitutive reference gene. ChIP experiments were performed in triplicates, and the data was normalized against the *35S::XAL1/xal1* control sample.

## 5. Conclusions

In this work, we reported that XAL1 induces flowering transition in response to a LD photoperiod or warm temperatures independently of CO and PIF4, respectively. *XAL1* is upregulated by light and temperature, probably via PHYB, and it participates in the same genetic pathway as FD under the LD photoperiod or independently of the latter at higher temperatures. XAL1 may fine-tune FT-derived regulation in response to those environmental signals by directly binding to *FD*, *SOC1*, *LFY*, and *AP1* promoters.

With the genetic and molecular results presented here, we propose that photoperiod and temperature signals derived from the leaves are partially independent of the ones directly transmitted in the shoot apical meristem to induce flowering. In this sense, PHYB has a dual function in flowering regulation, as a negative regulator that affects CO and PIF4 stability in the leaves and as a positive one that induces the expression of some flowering genes at the apical meristem.

Notably, a few degrees of variation between 22 and 27 °C change the genetic interaction between *XAL1* and *FD*. Furthermore, our results show that temperature induction is imposed over the photoperiod signaling in this genetic relationship and probably others. This confirms the importance of the effect of temperature on the interactions of the genetic network, and further studies need to be performed in the context of climate change.

## Figures and Tables

**Figure 1 ijms-24-12773-f001:**
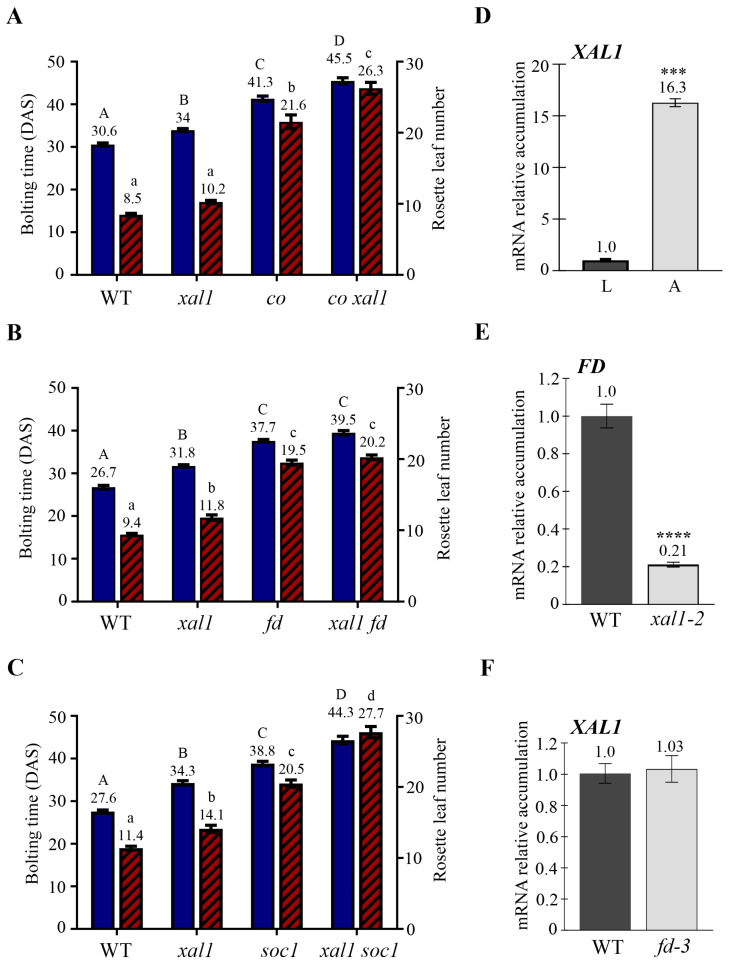
*XAL1* participates in the photoperiodic pathway by positively regulating *FD.* (**A**–**C**) Flowering time of *co-1 xal1-2* (**A**), *xal1-2 fd-3* (**B**), and *xal1-2 soc1-6* (**C**) compared to their parentals and WT plants grown under a LD photoperiod at 22 °C. Blue columns correspond to bolting time in Days After Sowing (DAS), and stripped red columns show the rosette leaf number. Statistical differences were obtained using one-way analysis of variance (ANOVA *p* < 0.01) followed by a Tukey’s multiple comparison test, indicated by capital letters (bolting time) and lowercase letters (rosette leaf number), n = 62–70 (**A**): 38 (**B**) and 21 (**C**) plants, respectively. (**D**) Relative accumulation of *XAL1* in WT leaves (L) and shoot apices (A). (**E**) Expression of *FD* in WT and *xal1-2* apices. (**F**) Accumulation of *XAL1* in *fd-3* apices, compared to WT. (**D**–**F**) 14-day-old soil-grown plants were used for RT-qPCRs analysis. Data represent the result of three biological replicates with two cDNAs for each genetic background. Statistical differences were obtained using a Student *T*-test *** *p* < 0.001, **** *p* < 0.0001. *PDF2*, *RNAH*, and *UPL7* genes were used as internal qPCR controls. Data represent mean value ± standard error.

**Figure 2 ijms-24-12773-f002:**
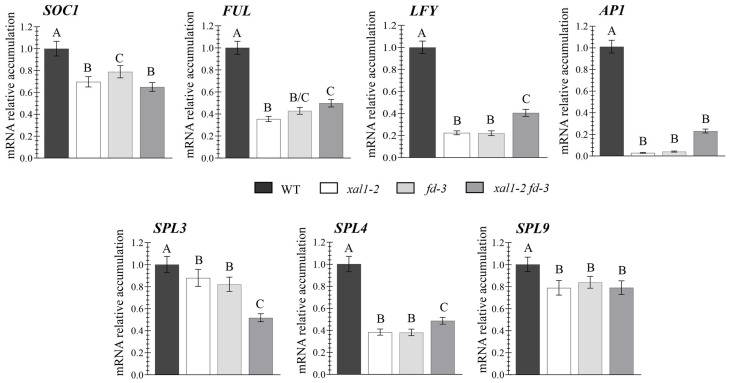
Regulation of different flowering genes by XAL1 and FD. Relative expression levels of *SOC1*, *FUL*, and *AP1*; the *SPL3*, *4*, and *9*; and *LFY* genes in the shoot apices of WT, *xal1-2*, *fd-3* and *xal1-2 fd-3*, plants grown under LD at 22 °C (14-day-old plants). Data represent the mean value ± se of two biological replicates with two cDNAs each. Statistical analysis was performed using one-way ANOVA (*p* < 0.001), followed by a Tukey’s multiple comparison test, denoted by capital letters. *PDF2* and *RNAH* were used as qPCR internal controls.

**Figure 3 ijms-24-12773-f003:**
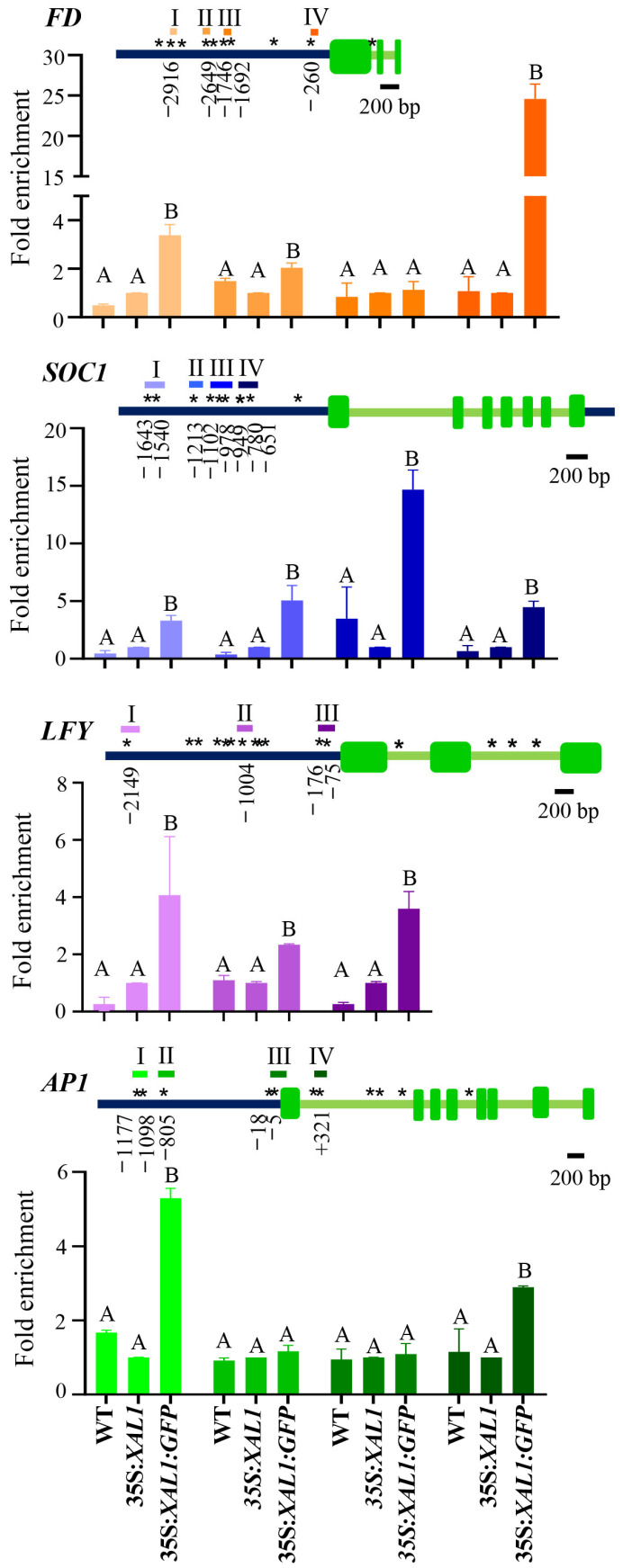
XAL1 binds to *FD*, *SOC1*, *LFY*, and *AP1* promoters. ChIP assays were performed using an anti-GFP antibody followed by qPCR amplifications of different DNA fragments (Roman numbers), from WT, *35S::XAL1/xal1-2* (without the GFP tag) and *35S::XAL1::GFP/xal1-2*. Fold enrichment is relative to the *35S::XAL1/xal1-2* control line for each DNA fragment and *ACTIN 2* was used as a constitutive reference gene. The data represent the average ± s.e. of three biological replicates. A one-way ANOVA test was performed followed by a Tukey’s multiple comparison test with a *p* < 0.001, except for fragments I and IV from *LFY* and *AP1*, respectively, where the differences correspond to a *p* < 0.05. Letters above the columns correspond to the results obtained by the statistical Tukey’s multiple comparison test, were B letters indicate significant differences against *35S::XAL1* (middle column) for each DNA fragment analyzed. CArG-boxes indicated by asterisks (*) were identified by PlantPAN (http://plantpan.itps.ncku.edu.tw/), New Place (https://www.dna.affrc.go.jp/PLACE/?action=newplace), Athamap (http://www.athamap.de/), and Meme suite 5.2 (https://meme-suite.org/meme/index.html) programs (accessed on 1 June 2022); the distances from the ATG of the motifs selected for theses assays are shown below each gene model. The scale bar represents 200 bp.

**Figure 4 ijms-24-12773-f004:**
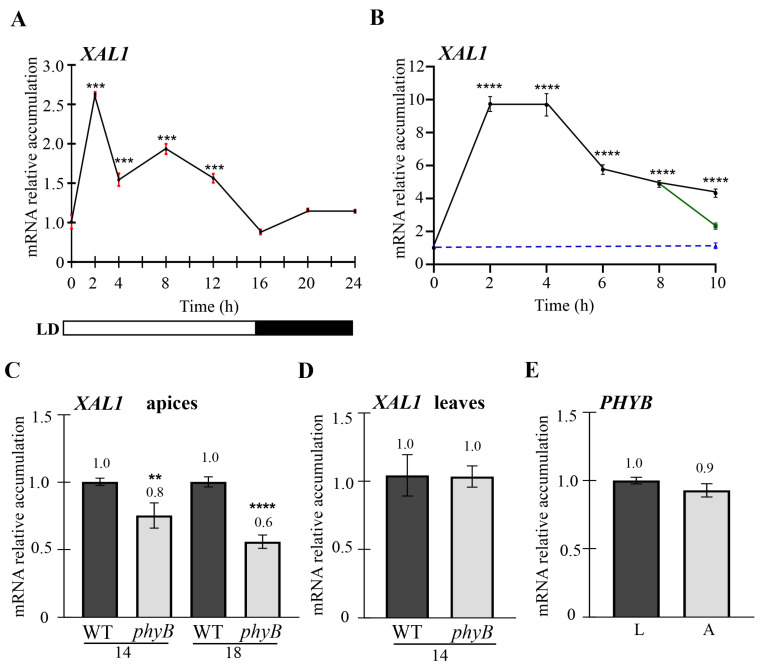
Light regulates *XAL1* via phytochrome B. (**A**) *XAL1* expression profile over 24 h in which zero corresponds to shoot apices collected 15 min before light exposure. (**B**) The *XAL1* expression profile on shoot apices collected every two hours of light exposure (black line). Plants were grown under LD for 13 days, followed by 2 days in the dark, and transferred back to light for 10 h. At the same time, part of the population remained in darkness (blue line), and another fraction was transferred to the dark for two hours after eight hours of light exposure (green line). (**C**) *XAL1* relative expression in *phyB* shoot apices compared to WT at 2 different ages (14 and 18 DAS). (**D**) *XAL1* mRNA accumulation in rosette leaves of 14-day-old WT and *phyB* plants. (**E**) Relative accumulation levels of *PHYB* in leaves (L) and apices (A) of WT plants grown under LD photoperiod. Data represent the mean value ± standard error. A total of 2 or 3 biological replicates were used with 27 plants each. *PDF2*, *RNAH*, and *UPL7* genes were used as qPCR internal controls. Statistical analysis was performed by a one-way ANOVA, followed by a Tukey’s multiple comparison test (**A**,**B**) or Student *T*-test (**C**–**E**): ** *p* < 0.01, *** *p* < 0.001, **** *p* < 0.0001.

**Figure 5 ijms-24-12773-f005:**
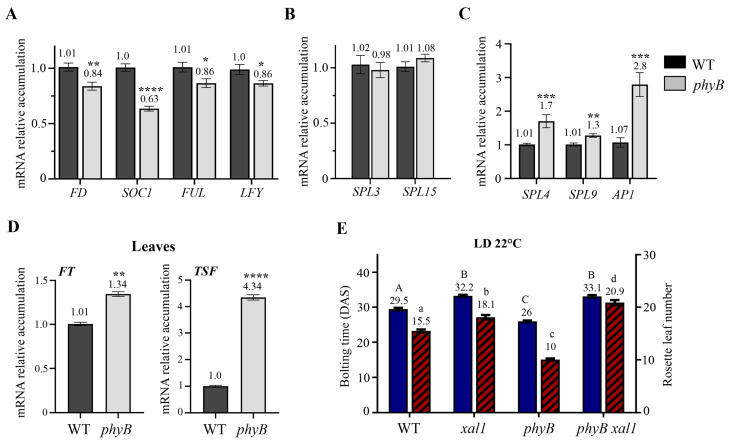
Phytochrome B differentially regulates flowering genes in the shoot apex and leaves under LD at 22 °C. (**A**–**C**) Expression of different flowering genes in shoot apices of 14 DAS *phyB* plants showing lower (**A**), no change (**B**), or increased (**C**) accumulation levels compared to the WT. (**D**) Induction of *FT* and *TSF* in *phyB* leaves compared to the WT. Statistical differences were determined using the Student *T*-test of three biological replicates (* *p* < 0.05, ** *p* < 0.01, *** *p* < 0.001 **** *p* < 0.0001). (**E**) Flowering time of *xal1-2 phyB* double mutant compared to single mutants and the WT. Statistical differences were obtained using a one-way ANOVA, followed by Tukey’s multiple comparison test *p* < 0.001 (n = 40 plants). Different letters denoted significant difference among genotypes. Capital letters correspond to bolting time and lower-case letters to the rosette leaf number. Data represent the mean value ± s.e.

**Figure 6 ijms-24-12773-f006:**
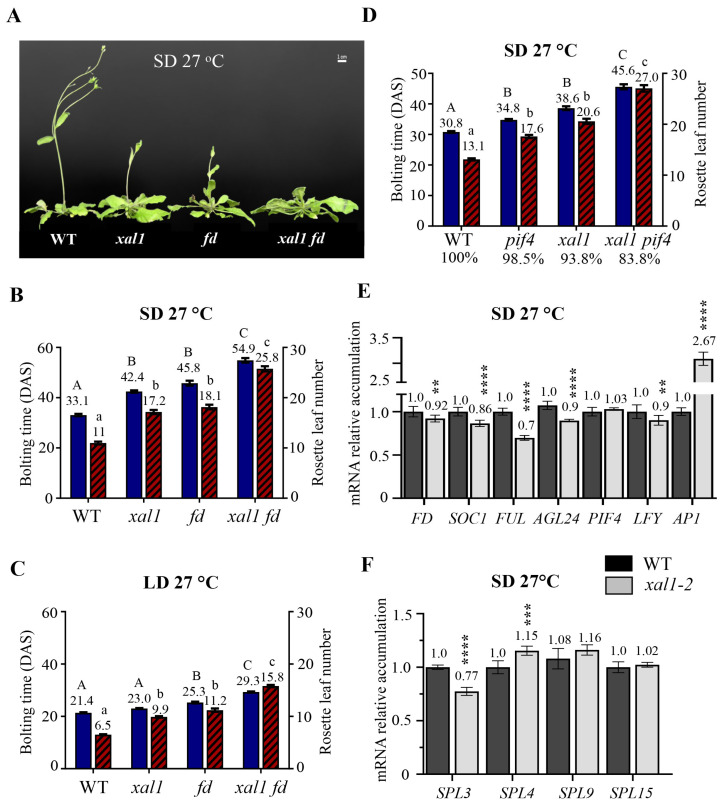
*XAL1* mediates the flowering transition by increased temperatures independently of *FD* and *PIF4*. (**A**–**C**) Flowering time of WT, *xal1-2*, *fd-3*, and *xal1-2 fd-3* grown at 27 °C under SD (**A**,**B**), or LD (**C**) photoperiods, respectively. Statistical differences were obtained using a one-way ANOVA followed by a Tukey’s multiple comparison test (SD n = 32–41; LD n = 54–78). (**D**) Flowering time comparison of the *xal1-2 pif4-2* double mutant with single mutants and WT plants under SD at 27 °C. The percentage of plants unable to flower over a 65-day period is indicated below each genotype. For statistical analysis, we performed a Kruskal–Wallis test followed by a Dunn’s multiple comparisons test (n = 56–68 plants). In (**B**–**D**), differences are indicated by capital letters (bolting time) and lowercase letters (rosette leaf number). (**E**,**F**) mRNA relative accumulation of several flowering genes in *xal1-2* apices compared to WT when plants are grown under an SD photoperiod at 27 °C. *RNAH* and *UPL7* genes were used as internal qPCR controls. Statistical differences were obtained using a Student *T*-test ** *p* < 0.01, *** *p* < 0.001, **** *p* < 0.0001. Data correspond to the mean value ± s.e.

## Data Availability

All data are reported in the article and the Appendix A.

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
