# Peer review of "XAANTAL1* Reveals an Additional Level of Flowering Regulation in the Shoot Apical Meristem in Response to Light and Increased Temperature in Arabidopsis"

_ijms, 2023, doi:10.3390/ijms241612773_

Round 1

Reviewer 1 Report

The authors presented an interesting topic and of high significance in the field.

Here are some comments that need to be addressed: 

Line 59: in the introduction part, why you used all Capital letters in expressing the name of the gene?

The same in line 73 and 77.

Figure1A, B, C: please explain what do you mean by the different letters on the figure as A a and B b. does this mean significance?

Figure2: please explain the meaning of all symbols in the figure as: I, II, III and 200pb …

Figure5: why you did not compare the mRNA expression levels in all treatments? You only used the wild type and the phyB mutants. Why did not compare the xal1 and the mixed mutant? 

Author Response

Answers Reviewer 1:

Line 59: in the introduction part, why you used all Capital letters in expressing the name of the gene?

Response: It is commonly known in the Arabidopsis nomenclature that genes and proteins when mentioned for the first time in the text, the full name is in capital letters, and the abbreviation goes in parentheses; for the rest of the text, only the abbreviation is required. This “rule” was applied to all the names of genes and proteins in the manuscript except for FD, which is the gene’s name per se. Furthermore, genes (capital letters) and mutants (lowercase) should be written in italics but not the proteins; thus, it is implicit in the mode it is written which one you are referring to. The article Meinke and Koornneef (1997). Community standards for Arabidopsis genetics. Plant J 12(2), 247-253, can be consulted in this regard.

The same in line 73 and 77.

R: already answered.

Figure1A, B, C: please explain what do you mean by the different letters on the figure as A a and B b. does this mean significance?

R: Yes, different letters mean significant differences among the genotypes. The difference between capital and lower-case letters refers to the parameter used for flowering time evaluation. As it says in the figure legend (lines 146-147): “Statistical differences were obtained by One-way analysis of variance (ANOVA P < 0.01) followed by a Tukey’s multiple comparison test, indicated by capital letters (bolting time) and lower-case letters (rosette leaf number),…”  

Figure2: please explain the meaning of all symbols in the figure as: I, II, III and 200pb …

R: I think you referred to Figure 3. The Roman numbers are related to the DNA fragments that were amplified. Lines 224-225 say, “ChIP assays were performed using an anti-GFP antibody followed by qPCR amplifications of different DNA fragments (Roman numbers), from WT, 35S::XAL1 (without the GFP tag) and 35S::XAL1::GFP”. For more clarity, we added the following sentence “B letters indicate significant differences against 35S::XAL1 (middle column) for each DNA fragment analyzed” (line 229). Other symbols like asterisks (line 230) and the numbers below gene models (line 233) were already explained in the figure legend. We meant to say bp, no pb, that was a mistake, and it has now been corrected in the figure. We also added “Scale bar represents 200 bp” at the end of the paragraph (line 233).

Figure5: why you did not compare the mRNA expression levels in all treatments? You only used the wild type and the phyB mutants. Why did not compare the xal1 and the mixed mutant? 

R: It was a question of timing. We first discovered that PHYB positively regulates XAL1, and then we analyzed the other flowering genes by RT-qPCR. After this analysis, we generated the xal1 phyB double mutant, and afterwards, we performed the flowering assay. Because we had some time limits to publish the data, and it was imperative to get the ChIP results to know if XAL1 could directly control some flowering genes, we concentrated our efforts on these experiments instead of collecting the material to repeat the RT-qPCRs with all the genotypes. This experiment would take 3-4 weeks, and we think it can be pursued in future studies and publications.

Reviewer 2 Report

Major points

1.     Line 181: Data about detection of OE line carrying a GFP tag should be shown.

2.     As for construction of OE lines it is confused with the experimental design. Are the 35S::XAL1::GFP construct were used to transform xal1-2 mutant but not WT plants? Besides the complementation of “short-root phenotype”, have the expression levels of Xal1 in these OE lines been confirmed compared to WT plant?   

3.     In Figure 4C, the data indicated that the expression levels of XAL1 in phyB mutant were decreased, however, as for the flowering phenotype (in Figure 5E), it is contradictory that phyB is early flowering, but the double mutant phyBxal1 is late flowering. Can the author better discuss about this point?

 Minor points

1.     In Figure 3, the legend for symbol representing scale should be “200 bp” but not “200 pb”.

2.     The reference gene used in ChIP assays (in Figure 3) should be mentioned in the corresponding figure legend or MM.

Minor editting for English will be needed.

Author Response

Answers to Reviewer 2:

  1. Line 181: Data about detection of OE line carrying a GFP tag should be shown.

Response: we agreed. We have included a confocal image in Figure S2B, and it is referred to in the text (line 181)

  1. As for construction of OE lines it is confused with the experimental design. Are the 35S::XAL1::GFP construct were used to transform xal1-2 mutant but not WT plants? Besides the complementation of “short-root phenotype”, have the expression levels of Xal1 in these OE lines been confirmed compared to WT plant?

R: We transformed both WT and xal1 mutant plants, but for this work, we only used the complementation lines. This is because only in the complementation lines we can be sure that XAL1 function properly. In some cases, it has been shown that overexpression of transcription factors can become deleterious. Therefore, we decided to use root-size complementation as a marker of the recovery of XAL1 functionality. Since the main purpose of generating these complementation lines was to use them in ChIP assays, it was essential to be sure that XAL1 could carry out its normal DNA binding. We have tried to clarify the text (lines 179-183). To answer about the expression, in Figure S2D it is clearly observed that XAL1 is constitutively overexpressed in the 35S:XAL1:GFP/xal1-2 line compared to WT at three different ages. However, to show a quantitative result, we have also included a RT-qPCR in which a high accumulation of XAL1 is observed in the 35S:XAL1:GFP/xal1-2 line compared to WT at 14 DAS (Figure S2E; lines 191-193).

  1. In Figure 4C, the data indicated that the expression levels of XAL1 in phyB mutant were decreased, however, as for the flowering phenotype (in Figure 5E), it is contradictory that phyB is early flowering, but the double mutant phyBxal1 is late flowering. Can the author better discuss about this point?

R: That is one of the innovative and surprising results of this work. Until now, it has been considered that PHYB in the leaf participates in CO degradation. Hence, in the phyB mutant, CO stabilizes and FT increases. This causes that FT travels to the meristem and induces early flowering. We discovered that PHYB at the apical meristem could regulate several flowering genes positively, despite its negative effect on CO. Thus, PHYB has a dual function, both positive and negative, and the regulation at the apex is partially independent of the signal derived from the leaves. Our data suggest that XAL1 is downstream from PHYB in the apices, and the upregulation of flowering genes mediated by both causes that when XAL1 and PHYB are removed, the double mutant is late flowering as xal1. In other words, the xal1 phenotype is imposed over the early flowering phenotype of phyB. We include in the discussion section 3.2, "one possible explanation is that in the absence of XAL1, downregulation of FD affects FT-FD dimmer formation, and this, combined with lower expression of other flowering genes positively regulated by FT-FD, PHYB and XAL1 at the apical meristem, is sufficient to prevent flowering, even when AP1 increases in the absence of PHYB” (lines 486-490).

More research is still needed, but these results will help to generate new hypotheses about the role of PHYB and XAL1 in regulating flowering. What is clear is that if the only function of PHYB were to promote CO degradation, the expected would have been that all the flowering genes at the apex would be induced in phyB, and this is not the case; only SPL4, SPL9 and AP1 were induced from the analyzed genes. Therefore, even when FT is a potent signal for flowering induction, it is not the only one.

The flowering gene regulatory network is vast (approximately 300 genes involved). Finding linear explanations is possible in one-one relationships, but it is too complicated to describe a phenotype. It will be important to study how PHYB functions, particularly at the meristem. Also, transcriptomic analysis and mathematical modelling will be needed to understand this apparent discrepancy.

Minor points

  1. In Figure 3, the legend for symbol representing scale should be “200 bp” but not “200 pb”.

R: That is correct, thank you. We have replaced the figure with that amendment.

  1. The reference gene used in ChIP assays (in Figure 3) should be mentioned in the corresponding figure legend or MM.

R: We agreed. We have included ACTIN 2 was used as a constitutive reference gene” in Figure 3 legend (line 227) and in methods (line 552).